# Nontuberculous Mycobacteria in Horses: A Narrative Review

**DOI:** 10.3390/vetsci10070442

**Published:** 2023-07-06

**Authors:** Lynna Li, Grazieli Maboni, Amy Lack, Diego E. Gomez

**Affiliations:** 1Department of Clinical Studies, Ontario Veterinary College, University of Guelph, Guelph, ON N1G 2W1, Canada; ll937@cornell.edu (L.L.); alack@uoguelph.ca (A.L.); 2Department of Population Health, College of Veterinary Medicine, University of Georgia, Athens, GA 30602, USA

**Keywords:** nontuberculous mycobacteria, zoonotic, gastroenteritis, placentitis, lymphadenitis

## Abstract

**Simple Summary:**

The incidence of diseases caused by nontuberculous mycobacteria (NTM) has increased in the past few years in both human and veterinary medicine. It was once thought that horses are immune to NTM infections, but there have been recent findings of NTM isolates within horses experiencing gastrointestinal, respiratory, and reproductive disorders. This literature describes the manifestations of NTM in horses, evaluates the efficacy of diagnostic and treatment methods, and discusses the zoonotic threat horses may pose in spreading NTM. Horses with NTM are most commonly presented for gastrointestinal infection, placentitis and abortion, respiratory distress, musculoskeletal abnormalities, and lymphadenitis. As of right now, genotyping diagnostic methods are the gold standard, but there has yet to be an effective treatment developed for NTM in both humans and horses. Lastly, the exact risk horses pose as a zoonotic threat to humans remains unclear, warranting further investigation.

**Abstract:**

Nontuberculous mycobacteria (NTM) infections are increasing in human and veterinary medicine. Although horses were initially thought to be resistant to NTM infection, reports of horses suffering from gastrointestinal, respiratory, and reproductive diseases associated with NTM have increased in the last few decades. The aim of this literature review is to summarize the mycobacteria species found in horses, describe clinical manifestations, diagnostic and treatment approaches, and public health concerns of NTM infection in horses. Clinical manifestations of NTM in horses include pulmonary disease, lymphadenitis, soft tissue, bone infections, and disseminated disease. NTM are also linked to granulomatous enteritis, placentitis, and abortions. Currently, diagnostic methods for NTM are limited and include acid-fast microscopy, bacterial cultures, species-specific PCR assays, and gene sequencing. In humans, NTM treatment guidelines are available, but their application appears inadequate and inconsistent. In horses, treatment guidelines for NTM infections are not available. NTM are a serious public health threat as 70% of people with untreated acquired immunodeficiency syndrome (AIDS) have a chronic pulmonary disease caused by NTM. Thus, it is essential that we gain a better understanding of NTM infections in horses and their zoonotic potential.

## 1. Introduction

During the past few decades, nontuberculous mycobacteria (NTM) have emerged as a pathogen causing human diseases, and there has been a perceived increase in the incidences of NTM in animals [1]. NTM are any type of mycobacteria that do not cause tuberculosis or leprosy, thus excluding the species of the *Mycobacterium tuberculosis* complex, *Mycobacterium leprae* and *Mycobacterium lepromatosis*. NTM are also commonly referred to as atypical mycobacteria, mycobacteria other than tuberculosis, or environmental mycobacteria. Horses are highly resistant to mycobacterial infections [2,3]. However, during the last few decades, the number of reports of horses suffering from gastrointestinal, respiratory, musculoskeletal, and reproductive diseases associated with infection with NTM have increased [4,5,6]. This literature review describes clinical features, diagnosis and treatment approaches, and public health considerations of NTM infection in horses. This review also summarizes the mycobacteria species that have been reported and associated with equine diseases so far.

## 2. Etiological Agents

Mycobacteria are members of the class Actinomycetia which includes bacteria known to cause respiratory diseases in horses (e.g., *Rhodococcus equi*), and placentitis and abortion (e.g., *Amycolatopsis* spp., *Streptomyces* spp. and *Crossiella equi*) [7]. Mycobacteria are members of the order *Actinomycetales* and the family *Mycobacteriaceae*. Mycobacteria are aerobic, non-spore-forming, non-motile, rod-shaped, acid-fast, and Gram-positive bacilli. The acid-fast feature is attributed to the rich cell wall composed of mycolic acids and complex lipids, allowing it to retain the carbol-fuchsin stain when decolorized with acid. To date, there are 188 species of mycobacteria with valid names. Recent studies based on 16S rRNA, 16S-23S spacer regions, multilocus and whole genome sequencing led to the division of the genus *Mycobacterium* into five genera, including *Mycobacterium*, *Mycolicibacter* gen. nov., *Mycolicibacillus* gen. nov., *Mycolicibacterium* gen. nov., and *Mycobacteroides* gen. nov. [8]. Based on growth rate, mycobacteria are often classified as fast growers (3–5 days, ex. *Mycolicibacterium smegmatis*) or slow growers (>7 days, ex. *Mycobacterium tuberculosis*, *M. tuberculosis* variant *bovis*, and *Mycobacterium avium*). Most of the pathogenic mycobacteria of horses and other animals are slow growers, which impairs laboratory diagnostics of clinical cases. Members of both tuberculous and nontuberculous mycobacteria groups have been associated with mycobacteriosis in horses. However, most of the infections are attributed to the NTM group (Table 1).

Most of the studies available in the literature indicate that both immunocompromised and non-immunocompromised horses are susceptible to several members of the NTM group (Table 1). Most of the pathogenic mycobacteria of horses belong to the *Mycobacterium avium* complex (MAC) which includes two species: *M. avium* and *M. intracellulare*. Four subspecies belong to the *M. avium* species, including *M. avium* subsp. *avium*, *M. avium* subsp. *hominissuis*, *M. avium* subsp. *silvaticum*, and *M. avium* subsp. *paratuberculosis* [9,10]. To date, the two MAC species and three subspecies of the *M. avium* are reported to cause mycobacteriosis in horses, including *M. avium* subsp. *avium*, *M. avium* subsp. *hominissuis*, and *M. avium* subsp. *silvaticum* (Table 1). *M. avium* subsp. *paratuberculosis* (MAP) is the agent of paratuberculosis disease affecting mostly cattle and other ruminants, but it has not been detected in spontaneous disease in domestic horses.

**Table 1 vetsci-10-00442-t001:** Summary of studies describing mycobacteria species associated with disease in horses.

Mycobacteria Species (Basonym)	Clinical Manifestation	Reference
*Mycobacterium terra*	Placentitis and abortion	[11]
MAC	Chronic weight loss	[12]
MAC	Pyrexia, weight loss, diarrhea, and dependent edema	[13]
MAC	Weight loss, bilateral mucoid nasal discharge, poor racing performance	[14]
MAC	Chronic lingual ulcer, intermittent diarrhea, weight loss, abortion, granulomatous colitis	[15]
MAC	Generalized dermatitis	[16]
MAC	Placentitis and abortion	[17]
MAC	Septic arthritis and granulomatous synovitis	[4]
*Mycobacterium avium* subsp. *avium*	Interstitial pneumonia	[2]
*Mycobacterium ulcerans*	Lichenification, hair loss, oedema, and ulceration on a fetlock, as well as a non-healing ulcer on the wither. Ulcerated lesion on its caudal thigh	[18]
*Mycobacterium avium* subsp. *hominissuis*	Horse 1: diarrhea and progressive weight loss. Nodular mass and thickened portion of the small intestineHorse 2: diarrhea, weight loss, and nasal discharge. Enlarged colonic lymph nodes with small nodular lesions, and corrugation and thickening of the colonic wall	[19]
*Mycobacterium avium* subsp. *silvaticum*	Diarrhea and weight loss. Nodular lesions on the peritoneal surface, the diaphragm, and the lymph nodes of the colon and cecum, as well as in the submucosa of the colon	[20]
MAC	Vaginal discharge followed by an abortion	[21]
*Mycobacterium avium strain 104*	Systemic infection in an aborted fetus	[22]
*Mycobacterium avium* subsp. *hominissuis*	Chronic diarrhea and weight loss. Pyrexia, lethargy, and inappetence	[6]
*Mycobacterium avium* subsp. *hominissuis*	Granulomas in the placenta and abortion	[23]
*Mycobacterium intracellulare*	Granulomatous rhinitis	[24]
*Mycobacterium branderi*	Chronic diarrhea, granulomatous mesenteric lymphadenitis	[5]

MAC, *Mycobacterium avium* complex.

## 3. Clinical Manifestation of NTM

Clinical manifestation of NTM in humans includes pulmonary disease (which comprised 90% of all NTM infections), lymphadenitis, soft tissue and bone infections, and disseminated disease [25]. In equids, infections with *M. avium* species are primarily associated with the gastrointestinal tract [12,13,26] and the reproductive system [11,15,17,21,22,23]. A limited number of equine cases included abnormalities of soft tissue [14,16], bone infection [4], and interstitial pneumonia [2]. In addition, a single case of generalized NTM infection has been reported in a horse from Norway [27].

### 3.1. Gastrointestinal Infection

Primary mycobacteriosis of the intestine is rarely reported in humans [28], although in patients with acquired immunodeficiency syndrome (AIDS), disseminated mycobacteriosis causing gastrointestinal disease has been reported. In horses, gastrointestinal mycobacteria infections are uncommon, but these bacteria have been linked to granulomatous enteritis and colitis in horses for several decades [6,12,15]. Granulomatous enteritis is an inflammatory infiltrative disease characterized by lymphoid and macrophage infiltration of the mucosal lamina propria with variable numbers of plasma cells and giant cells [29]. In 2015, nine horses (five Standardbred, two Finnhorse, one Finnish Warmblood, and one Norwegian Fjord horse) with disseminated alimentary mycobacteria infection associated (in five out of nine horses) with *M. avium* subsp. *hominissuis* were reported [6]. Seven of the nine horses were between 6 months and 2 years old, and two mature horses were also affected. The clinical signs of those horses were altered demeanor (i.e., obtundation), chronic diarrhea, weight loss, pyrexia, and ventral edema. Alterations in the complete blood cell count were unspecific, but biochemical abnormalities included hypoproteinemia characterized by hypoalbuminemia and hyperfibrinogenemia, which were present in all cases. Ultrasonographic examination of the abdomen revealed thickened small intestine in most horses. The duration of clinical signs before admission ranged from 2 weeks to 6 months.

The clinical signs reported in horses with disseminated alimentary mycobacteriosis are typically protein-losing enteropathies commonly associated with inflammatory bowel disease (IBD). In adult horses, IBD is documented as an abnormal cellular infiltration of the mucosa and submucosa of the gastrointestinal tract [30], including eosinophils, plasma cells, lymphocytes, basophils, or macrophages. Although an association between granulomatous enteritis in horses and the presence of *M. avium* existed, it was only in 2015 that NTM were also associated with lympho-plasmacytic infiltration of the small intestine [6]. The confirmatory test for infiltrative enteropathies is a full-thickness biopsy from the affected portion of the small intestine or colon obtained via exploratory surgery [30]. However, these procedures are invasive, and usually, they are not performed to diagnose chronic inflammatory enteritis. Thus, there is a possibility of underdiagnosis or misdiagnosis of NTM as a causative agent of lympho-plasmacytic ileitis and granulomatous enteritis in horses. Although there is no clear evidence that NTM are the causal agent of all granulomatous enteritis in horses, the study from Mönki J., et al. [6] provided compelling clinical, histopathological, and microbiological evidence that NTM, such as *M. avium* subsp. *hominissuis*, might be capable of inducing lympho-plasmocytic ileitis and granulomatous typhlocolitis in horses. Therefore, NTM should be considered a possible etiological agent in horses suspected of suffering chronic inflammatory infiltrative enteritis.

### 3.2. Placentitis and Abortion

During the last few decades, mycobacteria have been reported to cause placentitis and abortions in mares [11,15,17,21,22,23]. The first case of foal abortion associated with *Mycobacterium terra* infection was reported in 1981 [11]. In Japan, a 9-year-old thoroughbred mare was reported to have an infection with *Mycobacterium avium* leading to fetal mycobacteriosis and abortion [22]. Initially, the mare was examined due to a decreased estrogen level in the serum, but placentitis developed soon after the initial evaluation. The fetus was aborted 5 days later, and upon pathological examination, normal lymph node tissue was replaced by granulomatous lesions composed of macrophages, neutrophils, and multinucleated giant cells. A similar case was reported in the USA where a 25-year-old Thoroughbred mare was assessed at 5 months gestation for vaginal discharge. The mare was treated with altrenogest, trimethoprim sulfadiazine, and flunixin meglumine, but 2 days later, the mare aborted a nonviable fetus. Upon analysis, the fetus had a distended abdomen and yellow and friable liver, and the umbilical cord had fluid-filled sacculations associated with vessels in the allantois. Ziehl–Neelsen stain confirmed the presence of intracellular acid-fast bacilli in trophoblasts of the gravid horn and the cervical star area. A NTM (*Mycobacterium* Runyon group IV) was isolated from the chorioallantois and uterine fluid [21].

Recently, 10 cases of placentitis caused by *Mycobacterium avium* subsp. *hominissuis* infections in mares were reported in Japan [23]. The fetal ages at the time of the abortions ranged from 148 to 303 days. In most cases, the placenta had extensive pathological lesions, including white-yellow exudate on the surface. Histologically, mycobacterial granuloma formations were identified in the placenta and fetal organs. Bacterial culture of the placenta, uterus, and fetal organs (i.e., heart, lung, liver, kidney, spleen, and stomach contents) yielded the growth of acid-fast bacteria. Genome sequencing analysis indicated that the isolates were *Mycobacterium avium* subsp. *hominissuis*. Molecular analysis of the isolates showed that all patterns of the strains were identical, suggesting that the horses were infected with the same pathogen. Although mycobacteria are an uncommon cause of placentitis, mycobacteriosis should be suspected in mares with placental infections and abortions.

### 3.3. Respiratory Disease

Chronic pulmonary disease is the most frequent form of NTM in adult humans with MAC, *M. kansasii*, *M. abscessus*, and *M. fortuitum* being the most common NTM pulmonary pathogens in the United States [31,32]. In horses, a case of granulomatous rhinitis associated with infection with *M. intracellulare*, an uncommon species of MAC, was recently reported in Texas (USA) [24]. The horse was a 22-year-old Quarter Horse gelding with a 2-month history of unilateral mucopurulent nasal discharge. Radiographic examination and computerized tomography scan showed a soft tissue mass in the left rostral and caudal maxillary sinuses, left conchofrontal sinus, and left sphenopalatine sinus. The horse was diagnosed with granulomatous rhinitis, and histopathological examination of the mass revealed the presence of epithelioid macrophages, multinucleated giant cells, and plasma cells with multifocal areas of necrosis and degenerated neutrophils. On hematoxylin and eosin (HE) staining, faintly stained bacterial bacilli were visible within macrophages and multinucleated giant cells. In addition, numerous acid-fast bacilli were identified on acid-fast stain. The PCR amplification revealed that the pathogen shared 98% of identity with *M. intracellulare* [24]. This report indicates that NTM should be considered a differential diagnosis for horses with nasal masses and granulomatous rhinitis.

### 3.4. Lymphadenitis

Lymphatic diseases caused by NTM are common in children, and approximately 80% of the cases are caused by MAC [25]. NTM infection has also been reported as a causative agent of lymphadenitis in horses. *Mycobacterium branderi*, a NTM, was isolated from the mesenteric lymph nodes of a 17-year-old Thoroughbred stallion from Brazil with a 3-month history of chronic diarrhea, weight loss, depression, decreased appetite, fever, peripheral edema, and persistent leucocytosis [5]. Despite multiple courses of antimicrobial drugs, clinical signs persisted, and the horse was euthanized due to the grave prognosis. On post-mortem examination, most of the parenchyma of the mesenteric lymph nodes were replaced by inflammatory infiltration of macrophages, multinucleated giant cells, and some neutrophils, lymphocytes, and plasma cells. Numerous acid-fast bacilli within the cytoplasm of macrophages of mesenteric lymph nodes were observed on Ziehl–Neelsen stain [5]. Histopathological examination of the small and large intestinal sections revealed only lymphangiectasia and mucosal edema, and none of the sections contained any bacilli structures indicating the lesions were limited to the lymph nodes.

### 3.5. Musculoskeletal Disease

NTM are an uncommon cause of infection of the musculoskeletal system in humans [33]. In horses, MAC was cultured from the synovial fluid of a 12-year-old American Saddlebred gelding evaluated for chronic inflammation of the right radiocarpal joint (RCJ) [4]. The horse was treated for degenerative osteoarthritis with multiple injections of corticosteroids during the previous 36 months. Diagnostic imaging, including X-rays and computerized tomography, revealed a lytic lesion in the distal portion of the radius and periarticular bone proliferation at the right RCJ. Ultrasonographic examination showed a synovial proliferation of the RCJ. Biopsy samples from proliferative synovium were taken via arthroscopic joint examination and revealed proliferative granulomatous synovitis with the presence of giant cells. These findings indicated that NTM should be considered a differential diagnosis in animals with granulomatous synovitis.

## 4. Diagnosis of Nontuberculous Mycobacteria

### 4.1. Acid-Fast Microscopy

The diagnosis of NTM is challenging. Acid-fast microscopy is a rapid, simple, and economical method for quickly identifying subjects with mycobacterial infections. However, acid-fast microscopy cannot distinguish between mycobacteria species, or viable and nonviable bacteria. In addition, acid-fast microscopy cannot differentiate mycobacteria from other acid-fast positive bacteria such as *Nocardia* spp. or *Rhodococcus* spp. [34]. The sensitivity of acid-fast microscopy for diagnosing NTM infection can be affected by several factors, including the prevalence and severity of the disease, the type of sample analyzed, the quality and method of the sampling procedure, and the number of mycobacteria present in the sample [34]. The limitations of the acid-fast stain are also associated with the sampling procedure, quality and volume of the sample, number of bacteria in the tissue, manipulation of the sample, and expertise of the pathologist. For instance, in one study involving nine horses with alimentary mycobacteriosis, all rectal biopsies tested negative for acid-fast alcohol stain, whereas post-mortem examination revealed intracellular acid-fast bacilli in macrophages and multinucleated giant cells in the colon, liver, and lymph nodes in nine/nine horses and in the small intestine in five/nine horses [6]. However, the histopathologic findings of the rectum tissue on post-mortem examination were not reported, preventing the comparison of biopsies that usually contain only mucosa with full-thickness specimens collected on post-mortem examination. Therefore, future studies are warranted to evaluate the sensitivity of acid-fast microscopy of rectal biopsies in horses with mycobacteriosis of the gastrointestinal tract.

### 4.2. Culture

The isolation of NTM from clinical specimens can be laborious due to the potential overgrowth of coexisting non-acid-fast bacteria and the slow-growth nature of mycobacteria. The culture of NTM requires days to weeks before results are available. Multiple selective and differential media have been used for the isolation of mycobacteria. The Lowenstein–Jensen (L–J) medium and the Middlebrook 7H11 are commonly used to isolate mycobacteria from clinical specimens. Culture continues to be an essential method for the diagnosis of NTM. However, molecular methods are needed to distinguish mycobacteria genus, species, and subspecies that affect horses.

### 4.3. Molecular Methods

Genotyping methods have been successfully applied to assess the genetic divergence of NTM [34]. The most commonly used genotyping methods are pulsed-field gel electrophoresis, insertion sequence (IS)-based typing, methods based on minisatellite sequences, repetitive sequence-based PCR, random amplified polymorphic DNA analysis, amplified fragment length polymorphism analysis, or multilocus sequence typing. The selection of genotyping method greatly depends on the species, the sample, the purpose of the investigation, and the expected results. For instance, a previous study in horses achieved the discrimination between MAA and MAH using insertion sequence (IS)-based typing [6]. MAH isolates from horses included in that study were verified based on the absence of IS901 and the presence of IS1245 [35,36]. DNA sequencing of variable genomic regions offers a rapid, accurate method for identifying mycobacteria. Previously, the most regularly utilized and reliable approach to differentiate NTM was amplification and sequence analysis of hypervariable regions A and B of the gene encoding 16S rRNA [37]. Currently, sequencing of the heat shock protein gene (hsp65) and the RNA polymerase gene (rpoB) are considered the most reliable standard gene sequencing targets [22,38,39]. These methodologies can be used directly on the specimens collected from animals suspected of having mycobacterial disease. However, in some cases, these methods do not distinguish closely related NTM species, are time consuming, and require specialized and trained personnel.

## 5. Treatment

A significant concern in human medicine is the inadequate and inconsistent outcome of patients treated for NTM diseases [37]. The absence of antimicrobial agents with good in vivo activity against NTM and minimum adverse effects are the major limitations for the treatment of patients with NTM. Most first-line anti-tuberculosis drugs have 10 to 100 times less in vitro activity against NTM isolates than against *M. tuberculosis* [37]. The macrolides azithromycin and clarithromycin are two of the most active agents in treating NTM disease in humans. However, this therapy requires long-term administration, and more importantly, macrolide resistance develops with monotherapy, resulting in clinical and microbiological failure and relapse [40]. Thus, the most pressing questions are when to treat, what drugs to use, and how long the treatment should continue in horses with NTM infection. More importantly, should treatment be initiated without a confirmatory diagnosis? In human medicine, the American Thoracic Society and the Infectious Diseases Society of America guidelines recommend macrolides (i.e., clarithromycin or azithromycin), ethambutol, and rifamycin (e.g., rifampin or rifabutin). Anecdotal clinical experience in human medicine also supports the use of aminoglycoside therapy in widespread or refractory MAC infection [37]. The treatment protocol for NTM should be administered for 12 months after the infection site culture is negative, frequently resulting in regimens of 18 or more months [37]. A combination of macrolides with rifampin has been recommended to treat mycobacterial disease in horses [41]. This combination is commonly used for the treatment of infectious causes of IBD or pneumonia in young animals, including *Lawsonia intracellularis* [42,43] and *R. equi* [44]. Therefore, administering a combination of macrolide-rifampin could be indicated in horses with a definitive diagnosis of NTM. However, the length and antimicrobial combination for treating NTM infection in horses is currently unknown.

## 6. Public Health Concerns

During the past few decades, members of the NTM emerged as pathogens of human diseases, including lymphadenitis in children, pulmonary tuberculosis-like disease, and disseminated infections (occurring predominantly in immunocompromised humans, particularly those with acquired immunodeficiency syndrome) [25]. NTM have been identified in approximately 70% of subjects with advanced untreated AIDS and are considered the primary killer in this group of patients [32]. Notably, the incidences of chronic pulmonary disease caused by NTM in non-immunocompromised people have also been increasing worldwide [45,46]. One study in Ontario, Canada reported an increased trend in pulmonary NTM isolates from 1998 to 2010. Five-year prevalence increased from 29.3 cases per 100,000 persons from 1998 to 2002, to 41.3 per 100,000 from 2006 to 2010 [47]. In the United States, annual prevalence of NTM diseases in people aged 60 years or older increased from 19.6 per 100,000 from 1994 and 1996 to 26.7 per 100,000 from 2004 and 2006 [48]. Potential explanations for the increased incidence of NTM include increased environmental exposure to NTM from household water, more intensive antimicrobial drug usage creating NTM-permissive lung niches [49], and the spread of NTM through person-to-person transmission. Two recent studies investigating outbreaks in patients with cystic fibrosis using a whole-genome sequencing–based study have challenged the dogma of person-to-person transmission of NTM, indicating potential transmission between these patients [50,51]. Dogs can become infected by their owners, but the opposite situation, namely the dog as a source of infection for immunocompromised human beings, is the higher risk [52]. The exact route of NTM infection is not established with certainty; based on NTM environmental distribution, it appears that the organism is ingested, inhaled, or implanted [49]. In horses with alimentary mycobacterial infection, the exposure route appears to be oral–fecal contamination because of the predominant involvement of the gastrointestinal tract [6]. MAH is the most common NTM isolated from horses with alimentary mycobacteriosis [45]. Although these findings immediately increase the concern regarding the risk of human infection up to date, the shedding of mycobacteria in the feces of affected horses has not been reported. Further epidemiological and microbiology studies using whole-genome sequencing–based studies are warranted to determine whether horses could potentially be a source of NTM for humans.

## 7. Conclusions

NTM infections are emerging in human medicine. In horses, *Mycobacterium avium* complex is most implicated in NTM infections, and clinical manifestations include lymphadenitis, placentitis, and abortion. Acid-fast microscopy and culture are traditionally used for the diagnosis of NTM. However, genotyping methods have become more prevalent and accessible. NTM treatment guidelines are available in humans, but recommendations regarding treatment in horses are lacking. In addition, cases of NTM infections are increasing in both human and veterinary medicine. Thus, we must gain a better understanding of how these pathogens are transmitted among horses and determine the zoonotic potential.

## Data Availability

Not applicable.

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
