# Peer review of "Nontuberculous Mycobacteria in Horses: A Narrative Review"

_vetsci, 2023, doi:10.3390/vetsci10070442_

Round 1

Reviewer 1 Report

Overall, a good review of NTM infections; however, the authors need to mention the myriad diseases which are caused by bacteria within the Actinomyces genus.  These comprise a relatively small amount of the equine microbiome regardless of tissue examined; however, are the causative organism for many major diseases including Rhodococcus pneumonia (Prescotella equi) and other forms of placentitis, particularly Nocardioform placentitis (e.g. Amycolatopsis ssp, Crossiella equi, and Streptomyces).  These organisms are also acid-fast and simple culture/acid-fast staining cannot rule out these much more common causes of equine disease.  If the authors want to discuss the need to look for zebras, they cannot forget to look for the horses first. 

The authors did an excellent job pulling in the literature from equine NTMs.  I do believe a thorough review of other Actinomyces is beyond the scope of the study, but they do need to be mentioned, with some comparisons between them and NTMs, perhaps as a table comparing clinical/gross/histological signs. 

Minor comments: 

Don’t need MAC on every entry in Table 1 – define it in the legend and be done.  Still need to define the first instance in the text

Discuss acid-fast staining in the context of other Actinomyces bacteria

No major issues with the English 

Author Response

Reviewer 1

Overall, a good review of NTM infections; however, the authors need to mention the myriad diseases which are caused by bacteria within the Actinomyces genus.  These comprise a relatively small amount of the equine microbiome regardless of tissue examined; however, are the causative organism for many major diseases including Rhodococcus pneumonia (Prescotella equi) and other forms of placentitis, particularly Nocardioform placentitis (e.g. Amycolatopsis ssp, Crossiella equi, and Streptomyces).  These organisms are also acid-fast and simple culture/acid-fast staining cannot rule out these much more common causes of equine disease.  If the authors want to discuss the need to look for zebras, they cannot forget to look for the horses first.  

Response: Thanks for your comment we added a sentence in the etiologic agent section to acknowledge that Mycobacteria are members of the class Actinomycetia which includes bacteria known to cause respiratory diseases in horses (e.g., Rhodococcus equi), and placentitis and abortion (e.g., Amycolatopsis spp. Streptomyces spp. and Crossiella equi). Line 52 – 54.

The authors did an excellent job pulling in the literature from equine NTMs.  I do believe a thorough review of other Actinomyces is beyond the scope of the study, but they do need to be mentioned, with some comparisons between them and NTMs, perhaps as a table comparing clinical/gross/histological signs.  

Response: We agree with the reviewer that a thorough review of other Actinomyces is beyond the scope of this manuscript, however we added a sentence in the diagnosis section indicating that other acid-fast bacteria need to be differentiated from Mycobacterium. Line 207 – 208.

Minor comments:  

Don’t need MAC on every entry in Table 1 – define it in the legend and be done.  Still need to define the first instance in the text

Response: Table 1 was modified as suggested. MAC was defined at the first instance in the text. Line 73.

Discuss acid-fast staining in the context of other Actinomyces bacteria

Response: Added as suggested. Line 206 -208.

Reviewer 2 Report

General comments

The article presents an interesting and helpful review about the topic of nontuberculous mycobacterial infections in horses. The article is well written (some exceptions are addressed in specific comments); however, the authors should ensure that all abbreviations are introduced properly or an explanatory list of abbreviations is included.

The reviewer suggests including more specific information on sample collection in the section on diagnosis. Very little information is presented, and this would be an important topic for the equine practitioner if diagnostic testing of suspected infection is pursued.

The review contains information on human infections, and the public health aspect of human-horse interaction is touched upon. The reviewer suggests also including information – if available – on the potential for disease transmission between horses and other domestic or wild animals, as these may serve as important sources of infection for horses. Is there information on risk factors for equine infections? Is the role of immunocompromise similar to that described in humans?

Table 1 should be revised to be more readable. As specified in the specific comments, it is also confusing that the table appears to contain comprehensive information on the described equine cases, but is then followed by additional information in the text. It is unclear, why some references are summarized in the table and others are discussed in the text.  

Specific comments

Line 11/12: either “incidence has increased” or “diseases have become increasingly common”

Line 48: “infection with NTM”

Line 65: “pathogenic mycobacteria of horses” of “mycobacteria that are pathogenic to horses”

Line 72 ff: This paragraph is a bit confusing and difficult to follow, please consider re-phrasing.

Table 1: Formatting of the table is not ideal, especially the centralized column with clinical signs is hard to read and match up with the other columns. Please consult with the editor.

Line 85: “soft tissue and bone infections” (no comma)

Line 83 ff: Why are the clinical manifestations listed here not included in table 1? Table 1 gives the appearance of being an inclusive list of all reported cases in horses, such that it is confusing that additional descriptions of clinical manifestations are listed in the next paragraph.

Line 120: Do you mean “not until 2015” or “only in 2015”?

Line 128: Reference is variably indicated as “Monki et al” or “Mönki et al” (with Umlaut), please adapt.

Line 129: What is NTB? Is this supposed to read NTM? If not, please define the abbreviation.

Line 130: “typhlocolitis” is misspelled

Line 134: “have been reported”

Line 168: “infection with”

Line 212 ff.: For the diagnosis section, it would be very helpful to include information on adequate specimens for the different clinical syndromes related to NTM infections. Are fecal samples sufficient, for example, or are only tissue cultures useful in cases of IBD? (According to line 320, this appears not to be the case as organisms are not shed in feces) How about other easily collected specimens such as tracheal lavage samples, uterine swabs etc.?

Line 250: Do you really mean “rationality”? Should it rather read “purpose of the investigation” or similar?

Line 262: What does “it” (“it does not distinguish”) refer to? Based on the previous sentence, I would expect “they (the methods) do not distinguish…. are time consuming and require…..”.

Please see comments to the authors above

Author Response

General comments

The article presents an interesting and helpful review about the topic of nontuberculous mycobacterial infections in horses. The article is well written (some exceptions are addressed in specific comments); however, the authors should ensure that all abbreviations are introduced properly or an explanatory list of abbreviations is included. 

Response: Modified as suggested. Abbreviations were reviewed and introduced properly.

The reviewer suggests including more specific information on sample collection in the section on diagnosis. Very little information is presented, and this would be an important topic for the equine practitioner if diagnostic testing of suspected infection is pursued. 

Response: Modified as suggested throughout the manuscript.

The review contains information on human infections, and the public health aspect of human-horse interaction is touched upon. The reviewer suggests also including information – if available – on the potential for disease transmission between horses and other domestic or wild animals, as these may serve as important sources of infection for horses. Is there information on risk factors for equine infections? Is the role of immunocompromise similar to that described in humans? 

Response: Modified as suggested (see the response to specific comments)

Table 1 should be revised to be more readable. As specified in the specific comments, it is also confusing that the table appears to contain comprehensive information on the described equine cases, but is then followed by additional information in the text. It is unclear, why some references are summarized in the table and others are discussed in the text.  

Response: Modified as suggested (see the response to specific comments)

Specific comments

Line 11/12: either “incidence has increased” or “diseases have become increasingly common”

Response: Modified as suggested. Line 11 - 12.

Line 48: “infection with NTM”

Response: Modified as suggested. Line 47.

Line 65: “pathogenic mycobacteria of horses” of “mycobacteria that are pathogenic to horses”

Response: Modified as suggested. Line 66.

Line 72 ff: This paragraph is a bit confusing and difficult to follow, please consider re-phrasing. 

Response: Modified as suggested. Line 72 – 74.

Table 1: Formatting of the table is not ideal, especially the centralized column with clinical signs is hard to read and match up with the other columns. Please consult with the editor.

Response: Modified as suggested. Table 1.

Line 85: “soft tissue and bone infections” (no comma)

Response: Modified as suggested. Line 85.

Line 83 ff: Why are the clinical manifestations listed here not included in table 1? Table 1 gives the appearance of being an inclusive list of all reported cases in horses, such that it is confusing that additional descriptions of clinical manifestations are listed in the next paragraph. 

Response: Table 1 was modified as suggested.

Line 120: Do you mean “not until 2015” or “only in 2015”?

Response: Modified as suggested. Line 114.

Line 128: Reference is variably indicated as “Monki et al” or “Mönki et al” (with Umlaut), please adapt. 

Response: Modified throughout the manuscript.

Line 129: What is NTB? Is this supposed to read NTM? If not, please define the abbreviation. 

Response: good catch. It is indded NTM. Modified accordingly. Line 123.

Line 130: “typhlocolitis” is misspelled

Response: Modified as suggested. Line 124.

Line 134: “have been reported”

Response: Modified as suggested. Line 128.

Line 168: “infection with” 

Response: Modified as suggested. Line 160.

Line 212 ff.: For the diagnosis section, it would be very helpful to include information on adequate specimens for the different clinical syndromes related to NTM infections. Are fecal samples sufficient, for example, or are only tissue cultures useful in cases of IBD? (According to line 320, this appears not to be the case as organisms are not shed in feces) How about other easily collected specimens such as tracheal lavage samples, uterine swabs etc.?

Response: This information is presented in the section of clinical manifestation (highlighted in blue) and shows which specimens were used for the diagnosis of NTM in each of the cases. Line 91 – 201.

Line 250: Do you really mean “rationality”? Should it rather read “purpose of the investigation” or similar?

Response: Modified as suggested. Line 239.

Line 262: What does “it” (“it does not distinguish”) refer to? Based on the previous sentence, I would expect “they (the methods) do not distinguish…. are time consuming and require…..”.

Response: Modified as suggested. Line 250.

Reviewer 3 Report

Thank you for submitting your manuscript. I think it is an interesting review of an infectious agent not well studies in horses. 

Just one consideration about the Table 1: I would suggest (if possible based on the indications from the journal) to change the layout, as it is sometimes difficult to connect the clinical signs with the species of Mycobacterium it correlates to.

Author Response

Reviewer 3

Thank you for submitting your manuscript. I think it is an interesting review of an infectious agent not well studies in horses.

Response: Thanks for your comment  

Just one consideration about the Table 1: I would suggest (if possible, based on the indications from the journal) to change the layout, as it is sometimes difficult to connect the clinical signs with the species of Mycobacterium it correlates to.

Response: Table 1 was modified as suggested.